# Overlapping Pure LIVS Jr. Stents for Isolated Ruptured Dissecting Aneurysm of the Proximal Posterior Inferior Cerebellar Artery

**DOI:** 10.3390/medicina58020240

**Published:** 2022-02-05

**Authors:** Sungdae Lim, Kwangho Lee, Hyun Park, Won Heo, Soo-Hyun Hwang

**Affiliations:** 1Department of Neurosurgery, Gyeongsang National University School of Medicine and Gyeongsang National University Changwon Hospital, 11 Samjeongja-ro, Seongsan-gu, Changwon 51472, Korea; limsd87@naver.com (S.L.); wonni8@naver.com (W.H.); mdshhwang@naver.com (S.-H.H.); 2Department of Neurosurgery, Gyeongsang National University School of Medicine and Gyeongsang National University Hospital, 79 Gangnam-ro, Jinju 52727, Korea; 1coo3004@naver.com

**Keywords:** PICA, dissecting aneurysm, endovascular, subarachnoid hemorrhage, LIVS Jr.

## Abstract

We report our experience in treating a ruptured dissecting posterior inferior cerebellar artery (PICA) aneurysm. To our knowledge, this is the first reported case of overlapping stenting without coils for a ruptured dissecting aneurysm of the proximal PICA. A 66-year-old male patient presented with sudden altered mental state and a subarachnoid hemorrhage (SAH). The cerebral angiography revealed a long segmental dissecting aneurysm on proximal PICA. Overlapping stents were deployed to the dissecting site, and angiogram showed intact distal PICA flow and decreased contrast staining in the dissecting site. Successful flow diversion was achieved with stents. Procedure-associated complications did not occur. The patient’s postoperative course was uneventful. In follow-up cerebral angiography, dissecting aneurysm achieved complete remodeling. The decision that led to the choice of treatment is discussed.

## 1. Introduction

Isolated spontaneous dissecting aneurysms of the posterior inferior cerebellar artery (PICA) are rare lesions [1,2,3,4,5,6,7,8]. Recently, intracranial vessel dissections and dissecting aneurysms have been more frequently diagnosed due to advances in neuroimaging. However, management of PICA dissecting aneurysm remains controversial. Surgical treatments include direct proximal wrapping, clipping, and trapping with or without bypass [9]. However, manipulating the proximal PICA for clipping poses a high risk of postoperative neurologic morbidity because of the close relationships with the brain stem and lower cranial nerves. Endovascular treatment has been proposed as an alternative to surgery. Successful endovascular treatment of isolated dissecting aneurysms of the PICA has been reported, and most cases were performed by occlusion of the parent artery [4,8,9].

We report our experience with treatment of ruptured dissecting PICA aneurysm. To our knowledge, this is the first reported case of overlapping stenting without coils for isolated ruptured dissecting aneurysm of the proximal PICA.

## 2. Case Report

A 66-year-old male patient presented with sudden altered mental status. On arrival at our hospital his Glasgow Coma Score was 12/15. The patient’s past medical history included diabetes mellitus, hypertension and angina pectoris. He had taken non-vitamin-K antagonist oral anticoagulant (NOAC) for underlying disease.

The initial brain computed tomography (CT) scan revealed ventricular enlargement and diffuse subarachnoid hemorrhage at the perimedullary area with IVH at the third and fourth ventricle. For a differential diagnosis, CT angiography was performed and revealed a pearl and string pattern at the right proximal PICA. Then, the drainage of cerebrospinal fluid (CSF) was performed through external ventricular drainage via the Rt. Kocher’s point. Immediately after, cerebral angiography was performed. The right vertebral arterial (VA) angiogram revealed an approximately 10 mm-long segmental dissecting aneurysm on proximal PICA and large PICA vessels with PICA dominance over anterior inferior cerebellum artery (AICA) in supply of the cerebellum (Figure 1). On external carotid angiogram, the occipital artery (OA) was hypoplastic. Although there was a contralateral PICA, its size was too small to expect collateral circulation with extracranial PICA (Figure 1B).

Under general anesthesia, endovascular treatment was performed. Vascular access was obtained via the right common femoral artery. A 6-French guide catheter was placed in the right proximal VA, and a microcatheter was advanced distal to the tonsilomedullary segment of the PICA under guided microwire. An LIVS Jr. (MicroVention, Tustin, CA, USA) woven stent was deployed to cover the entire dissection length to save the flow within the PICA. After 30 min, angiogram revealed inadequately decreased contrast staining in the dissecting site. Therefore, a second LIVS Jr. stent was deployed to overlap the stent using in-stent navigation by microwire. At 30 min after second stenting, angiogram showed an intact distal PICA flow and more decreased contrast staining in the dissecting site (Figure 2). Successful flow diversion was achieved with stents. Procedure-associated complications did not occur.

The patient’s postoperative course was uneventful. For 3 weeks, cerebral angiography and magnetic resonance (MR) imaging were performed. No evidence of infarction or stenotic lesion were identified upon examinations (Figure 3A). At discharge, the patient had a modified Rankin scale grade of 1 with mild weakness of both legs. Eight months later, in his follow-up angiography, the dissecting aneurysm had developed complete remodeling (Figure 3B,C). During follow-up, the patient had recovered completely with no neurologic deficits.

## 3. Discussion

PICA dissecting aneurysms account for 0.5~0.7% of all intracranial aneurysms [10]. Acute posterior circulation dissections carry high rebleeding risk (24%) with an associated high mortality rate [11]. Clinical presentations include symptoms of subarachnoid hemorrhage and/or ischemic symptoms. SAH has been reported as the most common initial clinical presentation (74%) [7]. The cause of PICA dissection is unknown. Although several conditions have been associated with the dissections, no associating factors have been clearly linked with isolated PICA dissections [7]. Although CT and MR angiography have been useful, cerebral angiography is the gold standard for the diagnosis of dissecting aneurysm. On angiography, dissection may appear as a string and pearl sign, segmental narrowing, occlusion of the vessel, or the pathognomonic double lumen.

Numerous vascular anatomic variations in VA and PICA with perforating vessels make a standardized approach difficult. Therefore, the management of PICA dissections remains controversial and challenging. Treatment of PICA dissection includes medical, surgical, and endovascular options. For patients who present with a PICA infarct, conservative treatment can be recommended if there are no obvious angiographic risk factors for hemorrhage, such as a pseudoaneurysm. For patients who are treated with anticoagulation, Tawk et al. recommend close follow-up and a repeat angiogram within 2 weeks of presentation [7]. Only patients who do not present with hemorrhage are good candidates for medical treatment. Patients with a hemorrhagic presentation or worsening dissection need to invasive therapy such as surgical or endovascular treatment.

Surgical treatment includes various options such as proximal clipping, trapping, and wrapping or resection with PICA end-to-end anastomosis, PICA-to-PICA side-to-side anastomosis, OA-to-PICA anastomosis or PICA reimplantation to the VA [2,5,7,10,12,13,14,15]. However, surgical management carries a high risk of postoperative neurological morbidity because of the close relationship between the brain stem and lower cranial nerves [14,16]. Horowitz et al. reported 66% postoperative neurologic morbidity [14]. In this case, the patient had a long segment dissecting aneurysm, hypoplastic OA. Thus, the patient was not a suitable recipient for bypass surgery on cerebral angiography.

The advantages of endovascular management include the avoidance of manipulation of the brain stem, cerebellar tissue and lower cranial nerves. Most of the endovascular treatments of PICA dissecting aneurysms have included proximal occlusion [4,17,18]. Although patients treated with proximal occlusion had a decreased risk of rebleeding, the possibility of occlusion of perforators as well as infarction in the territory of PICA still remains. Lister et al. studied PICA anatomy in 25 adult cadavers. They reported that the perforating branches emanated from the first three segments of the PICA (anterior medullary, lateral medullary, tonsillomedullary) and especially from the first one [19].

In some cases, flow-diverting stent (FDS) treatment was used in the dissecting aneurysm to produce good results [20]. However, in other cases, FDS is not available or is difficult to use. In this case, there is a possibility that the use of a larger microcatheter may further aggravate the dissecting site with the advancement of the microcatheter, because the microcatheter to use a FDS is more rigid and larger. Therefore, the use of the smallest currently available microcatheter was considered to be safe. For this reason, overlapping stents were implemented. In a computational fluid dynamic study, the double-LVIS stent resulted in a better flow-diverting effect than a Pipeline device [21].

We have not tried the preventive occlusion test. Affected PICA occlusion was expected to result in large infarction of the cerebellum due to the small size of both AICA and contralateral PICA. Therefore, proximal PICA occlusion was considered to be the last treatment option. Thus, if the pseudoaneurysm is located distally, a sacrifice of the PICA distal to its third segment (approximately halfway between the caudal and rostral loops of the PICA) will be usually well-tolerated. Since no data are available on the angiographic signs to predict whether occlusion of the PICA would be tolerated, a preventive test occlusion is usually recommended [4]. However, this test does not prevent occlusion of perforators, because perforators cannot be easily seen in angiography.

In our patient, the dissecting aneurysm involved a relatively straight, long segment of the proximal PICA. Surgical treatment was difficult because of the patient’s premedical history, the long segment involvement, hypoplastic OA, and hypoplastic contralateral PICA as well as his being unsuitable for bypass. We had decided that deployment of overlapping stents using a flow-diverting effect in the dissecting site would result in a good outcome. This strategy did not completely eliminate the potential for risk of rebleeding; however, in the situation of an unestablished treatment of PICA dissecting aneurysm, overlapping stenting without coils can be considered as another option.

## 4. Conclusions

Optimal treatment strategies for dissecting aneurysms of PICA have not yet been established. However, in cases where bypass surgery was unsuitable or in an expected large-territory infarction after occlusion of the parent artery, overlapping stenting without coils may be a useful option for straightened long-segment-involved dissecting aneurysm of proximal PICA.

## Figures and Tables

**Figure 1 medicina-58-00240-f001:**
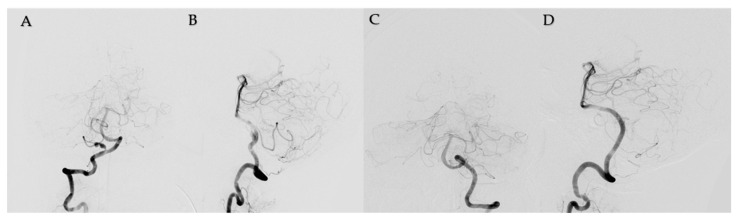
The right vertebral arterial angiogram (**A**,**B**) showed an approximately 10 mm-sized long segmental dissecting aneurysm on proximal posterior inferior cerebellar artery (PICA) and large PICA vessels with PICA dominance over the anterior inferior cerebellum artery in supply of the cerebellum. Left vertebral arterial angiogram was shown. (**C**,**D**).

**Figure 2 medicina-58-00240-f002:**
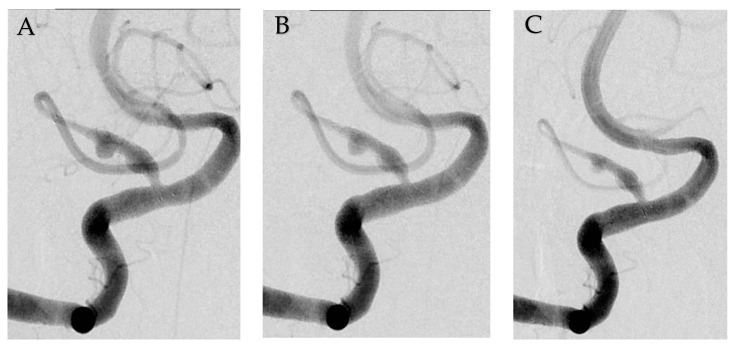
Serial angiogram showing flow diverting effect. Angiogram showed posterior inferior cerebellar artery after single LIVS Jr. 2.5x23 stent deployment (**A**). Immediate angiogram after second LVIS Jr., 2.5x17 stent deployment (**B**). 30 min later, angiogram demonstrated a decreased contrast filling in the dissecting aneurysm (**C**).

**Figure 3 medicina-58-00240-f003:**
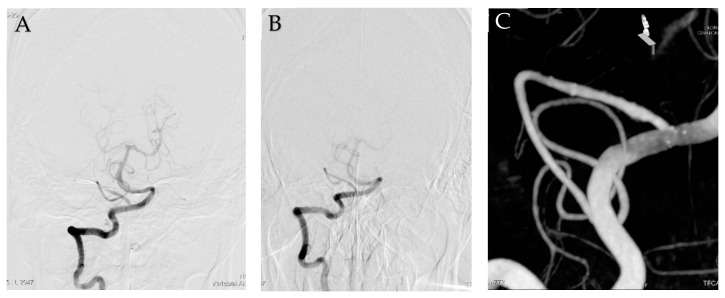
Follow-up cerebral angiography. Angiography showed an invisible dissecting aneurysm at 3 weeks later (**A**). Eight months later, Angiogram (**B**) and reconstructive image (**C**) showed complete remodeling of dissecting aneurysm.

## Data Availability

Data available in a publicly accessible repository.

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
