# Peer review of "Overlapping Pure LIVS Jr. Stents for Isolated Ruptured Dissecting Aneurysm of the Proximal Posterior Inferior Cerebellar Artery"

_medicina, 2022, doi:10.3390/medicina58020240_

Round 1

Reviewer 1 Report

For the proximal dissecting aneurysm of PICA, the endovascular treatment was difficult. We did not know what intervention was right.

I have some concerns.

  1. Why the dissection was treated with two overlapping LVIS, not one? What was the evidence of two LVIS?
  2. The tail of LVIS was in the middle of vertebral artery, it can disturb the blood flow, which should be considered.
  3. In general, the dissection was extensive like this case, no perforators were from the PICA to supply the brain stem, so, the parent artery occlusion can be considered after the balloon occlusion test. Please add the information.
  4. What about the contralateral PICA? Did the collateral circulation see?
  5. Why did the flow diverting stent consider?

Reviewer 2 Report

Authors herein present a case of a SAH patient with a dissecting aneurysm of a PICA arising from a dominant VA. The patient was successfully treated with a double stenting endovascular technique. Pre,intra and postprocedural images of angiograms are provided confirming the initial success of the procedure and a short follow up note of the evolution of the patients is disclosed. Discussion concisely elaborates on the main potential points of argument that should be considered on the decission-making of such a case.

Overall, this manuscript is a well described case report, however there are various articles reporting the single-center experience of some endovascular teams treating similar aneurysms with comparable devices. (The use of flow diverters to treat aneurysms of the posterior inferior cerebellar artery: Report of three cases. Bhogal et al.).

The novelty of using two stents with a flow diverting effect instead of a FDD may not be well justified or may not be enough to endorse this article for publication.

Minor corrections:

  • Moderate english language review.
  • Consider changing OA-to-PICA instead PICA-to-OA in order to clearly determine donor-to-recipient flow direction.
  • Conclusion should not include references. Consider transferring 135-143 lines to the discussion.

Author Response

Reviewer #2
Point 1: Authors herein present a case of a SAH patient with a dissecting aneurysm of a PICA arising from a dominant VA. The patient was successfully treated with a double stenting endovascular technique. Pre,intra and postprocedural images of angiograms are provided confirming the initial success of the procedure and a short follow up note of the evolution of the patients is disclosed. Discussion concisely elaborates on the main potential points of argument that should be considered on the decission-making of such a case.
Overall, this manuscript is a well described case report, however there are various articles reporting the single-center experience of some endovascular teams treating similar aneurysms with comparable devices. (The use of flow diverters to treat aneurysms of the posterior inferior cerebellar artery: Report of three cases. Bhogal et al.).
Point 2: Minor corrections:
Moderate english language review.
Consider changing OA-to-PICA instead PICA-to-OA in order to clearly determine donor-to-recipient flow direction.
Conclusion should not include references. Consider transferring 135-143 lines to the discussion.
Response
Response 1: Thankfully, I read the paper you presented using Flow Diverter stent well. Our case is believed to be a therapeutic example if FDS is not available or difficult to use in the treatment of similar delaminated cerebral aneurysms
Response 2: change OA-to-PICA and transfer 135-143 lines to the discussion.
As the reviewer’s recommendation, we described it on Discussion and conclusion parts as follows :
Discussion
PICA dissecting aneurysms account for 0.5~0.7% of all intracranial aneurysms [10]. Acute posterior circulation dissections carry high rebleeding risk (24%) with an associated high mortality rate [11]. Clinical presentations include symptoms of subarachnoid hemorrhage and/or ischemic symptoms. SAH has been reported as the most common initial clinical presentation (74%) [7]. The cause of PICA dissection is unknown. Although several conditions have been associated with the dissections, no associating factors have been clearly linked with isolated PICA dissections [7]. Although CT and MR angiography has been useful, cerebral angiography is the gold standard for the diagnosis of dissecting aneurysm. On angiography, dissection may appear as a string and pearl sign, segmental narrowing, occlusion of the vessel, or the pathognomonic double lumen.
Numerous vascular anatomic variations in VA and PICA with perforating vessels make a standardized approach difficult. Therefore, the management of PICA dissections remains controversial and challenging. Treatment of PICA dissection includes medical, surgical, and endovascular options. For patients who present with a PICA infarct, conservative treatment can be recommended if there are no obvious angiographic risk factors for hemorrhage such as a pseudoaneurysm. For patients who are treated with anticoagulation, Tawk et al.
recommend close follow-up and a repeat angiogram within 2 weeks of presentation [7]. Only patients who do not present with hemorrhage are good candidates for medical treatment. Patients with a hemorrhagic presentation or worsening dissection need to invasive therapy such as surgical or endovascular treatment.
Surgical treatment includes various options such as proximal clipping, trapping, and wrapping or resection with PICA end-to-end anastomosis, PICA to PICA side-to-side anastomosis, OA-to-PICA anastomosis or PICA reimplantation to the VA [2, 5, 7, 10, 12-15]. However, surgical management carries a high risk of postoperative neurological morbidity because of the close relationship between the brain stem and lower cranial nerves [14,16]. Horowitz et al reported 66% postoperative neurologic morbidity [14]. In this case, the patient had a long segment dissecting aneurysm, hypoplastic OA. Thus, the patient was not a suitable recipient for bypass surgery on cerebral angiography.
The advantages of endovascular management include the avoidance of manipulation of the brain stem, cerebellar tissue and lower cranial nerves. Most of the endovascular treatment of PICA dissecting aneurysms had included proximal occlusion [4, 17, 18]. Although patients treated with proximal occlusion had a decreased risk of rebleeding, the possibility of occlusion of perforators as well as infarction in the territory of PICA still remains. Lister et al. studied PICA anatomy in 25 adult cadavers. They reported that the perforating branches emanated from the first three segments of the PICA (anterior medullary, lateral medullary, tonsillomedullary) and especially from the first one [19].
In some cases, flow diverting stent(FDS) treatment was used in the dissecting aneurysm to produce good results[20]. But, in other cases, FDS is not available or difficult to use. In this case, there is a possibility that the use of a larger microcatheter may further aggravate
dissecting site when the advancement of microcatheter because a microcatheter to use a FDS is more rigid and larger. Therefore, the use of the smallest currently available microcatheter was considered to be safe. For this reason, overlapping stents were implemented. In a computational fluid dynamic study, the double-LVIS stent resulted in a better flow diverting effect than a Pipeline device [21].
We have not tried the preventive occlusion test. Affected PICA occlusion was expected to result in large infarction of the cerebellum due to the small size of both AICA, contralateral PICA. Therefore, proximal PICA occlusion was considered to last treatment option. Thus, if the pseudoaneurysm is located distally, a sacrifice of the PICA distal to its third segment (approximately halfway between the caudal and rostral loops of the PICA) will be usually well tolerated. Since no data are available on the angiographic signs to predict whether occlusion of the PICA would be tolerated, a preventive test occlusion is usually recommended [4]. However, this test does not prevent occlusion of perforators, because perforators cannot be easily seen in angiography.
In our patient, the dissecting aneurysm involved a relatively straight long segment of the proximal PICA. Surgical treatment was difficult because of the patient’s premedical history, long segment involvement, hypoplastic OA, and hypoplastic contralateral PICA as well as his being unsuitable for bypass. We had decided deployment of overlapping stents using a flow diverting effect in the dissecting site would result in a good outcome. This strategy did not completely eliminate the potential for risk of rebleeding; however, in the situation of an unestablished treatment of PICA dissecting aneurysm, overlapping stenting without coils can be considered as another option.
Conclusions
Optimal treatment strategies for dissecting aneurysm of PICA have not yet been established. However, in cases where bypass surgery was unsuitable or in an expected large territory infarction after occlusion of the parent artery, overlapping stenting without coils may be a useful option for straightened long segment-involved dissecting aneurysm of proximal PICA.
Thank you very much again for your review.